# Predicting Invasiveness in Lepidic Pattern Adenocarcinoma of Lung: Analysis of Visual Semantic and Radiomic Features

**DOI:** 10.3390/medsci12040057

**Published:** 2024-10-18

**Authors:** Sean F. Johnson, Seyed Mohammad Hossein Tabatabaei, Grace Hyun J. Kim, Bianca E. Villegas, Matthew Brown, Scott Genshaft, Robert D. Suh, Igor Barjaktarevic, William Dean Wallace, Fereidoun Abtin

**Affiliations:** 1Department of Radiological Sciences, David Geffen School of Medicine, University of California Los Angeles, Los Angeles, CA 90024, USAbevillegas@mednet.ucla.edu (B.E.V.);; 2Department of Radiology, Massachusetts General Hospital, Harvard Medical School, Boston, MA 02114, USA; 3Department of Pulmonary and Critical Care Medicine, David Geffen School of Medicine, University of California Los Angeles, Los Angeles, CA 90024, USA; 4Department of Pathology, Keck School of Medicine, University of Southern California, Los Angeles, CA 90089, USA

**Keywords:** lung biopsy, invasiveness prediction, lepidic predominant adenocarcinoma, semantic features, radiomic features

## Abstract

Objectives: To differentiate invasive lepidic predominant adenocarcinoma (iLPA) from adenocarcinoma in situ (AIS)/minimally invasive adenocarcinoma (MIA) of lung utilizing visual semantic and computer-aided detection (CAD)-based texture features on subjects initially diagnosed as AIS or MIA with CT-guided biopsy. Materials and Methods: From 2011 to 2017, all patients with CT-guided biopsy results of AIS or MIA who subsequently underwent resection were identified. CT scan before the biopsy was used to assess visual semantic and CAD texture features, totaling 23 semantic and 95 CAD-based quantitative texture variables. The least absolute shrinkage and selection operator (LASSO) method or forward selection was used to select the most predictive feature and combination of semantic and texture features for detection of invasive lung adenocarcinoma. Results: Among the 33 core needle-biopsied patients with AIS/MIA pathology, 24 (72.7%) had invasive LPA and 9 (27.3%) had AIS/MIA on resection. On CT, visual semantic features included 21 (63.6%) part-solid, 5 (15.2%) pure ground glass, and 7 (21.2%) solid nodules. LASSO selected seven variables for the model, but all were not statistically significant. “Volume” was found to be statistically significant when assessing the correlation between independent variables using the backward selection technique. The LASSO selected “tumor_Perc95”, “nodule surround”, “small cyst-like spaces”, and “volume” when assessing the correlation between independent variables. Conclusions: Lung biopsy results showing noninvasive LPA underestimate invasiveness. Although statistically non-significant, some semantic features showed potential for predicting invasiveness, with septal stretching absent in all noninvasive cases, and solid consistency present in a significant portion of invasive cases.

## 1. Introduction

Lung cancer remains the leading cause of cancer-related death and the second most diagnosed cancer in the United States [1]. Adenocarcinoma is the most common type of primary lung cancer [2]. Adenocarcinoma with a lepidic growth pattern is characterized by tumor cells proliferating along the surface of intact alveolar walls without stromal or vascular invasion by pathologic assessment [3].

The International Association for the Study of Lung Cancer/American Thoracic Society/European Respiratory Society (IASLC/ATS/ERS) classification criteria [4] for subtyping of adenocarcinoma with lepidic growth pattern is based on size of tumor and presence or absence of stromal invasion. This includes adenocarcinoma in situ (AIS) which has an entirely lepidic growth pattern and is >0.5 cm and ≤3 cm in size, minimally invasive adenocarcinoma (MIA) if ≤3 cm with ≤0.5 cm invasion, and lepidic predominant adenocarcinoma (LPA) if a tumor with a lepidic-predominant growth pattern is >3 cm, has >0.5 cm invasion, or has lymphovascular/pleural invasion of any size. AIS and MIA have been shown to have favorable prognosis with close to 100% 5-year cancer-specific survival rate compared to LPA with a cancer-specific survival rate between 85.7 to 100% post-resection [5].

A previous study by Young et al. demonstrated that there were no statistically significant differences in sex and age between AIS/MIA disease and iLPA. A number of demographic factors have been shown to be associated with invasive disease. For example, race was associated with invasive disease, specifically Asian descent. Smoking history was shown to be inversely associated with invasive disease whereas active smoking and pack years were not. A history of extrathoracic cancer less than 5 years prior to biopsy of the lung nodule was also associated with invasive disease [6]. However, the study did not evaluate the visual or texture features of the nodules biopsied to predict invasive disease.

Manifestations of LPA on CT can vary, but usually it appears as a part-solid nodule or mass [4,7]. The ground glass component of the lesions has been shown to correlate with a lepidic growth pattern on pathology [7]. Performing percutaneous needle biopsy of the lung under computed tomography guidance is essential for evaluating pulmonary abnormalities, given its exceptional diagnostic accuracy of around 93% in detecting malignancies [8]. After biopsy confirmation, one proposed surgical approach to management of lepidic predominant-growth pattern lesions has been a sublobar resection as opposed to a lobectomy, which is recommended for histologically more aggressive (non-lepidic) lung adenocarcinoma of similar size. Patients who have AIS and MIA stand to benefit from sublobar resection, but there is an increased risk of recurrence if the adenocarcinoma is invasive lepidic pattern adenocarcinoma (iLPA) [9,10,11]. To confirm the diagnosis of AIS or MIA, however, an evaluation of the tumor post-resection is necessary to exclude invasive disease.

In cases where biopsy results only show a noninvasive lepidic tumor, treatment often becomes delayed, with a false sense of security for the treating physician and patient, and the patient may need to undergo complete lobectomy at a later stage. Since the surgical approach and outcome differ between noninvasive lepidic tumors and LPA, it is important to predict iLPA despite a biopsy revealing only noninvasive lepidic growth.

This study aims to test various imaging parameters to predict iLPA tumors in biopsy-proven AIS/MIA using visual semantic and quantitative CAD-based texture analysis (CBTA).

## 2. Materials and Methods

This study is a retrospective analysis of a lung nodule biopsy database from a single academic center. It was approved by the University of California Los Angeles (UCLA) Institutional Review Board (IRB#17-001536). Since this study was retrospective and all the data and images used were fully anonymized, the requirement of informed consent was waived.

### 2.1. Subject Selection

Review of pathology database of subjects who were 18 years and older and underwent CT-guided core needle biopsy (CNB) which revealed a diagnosis of noninvasive LPA and subsequently underwent surgical resection from 2011 to 2017 were identified. Pathology assessment was performed as a standard of care. For the purpose of this study, all pathology slides were reassessed by a chest pathologist with 20 years of experience (WDW) to confirm the original diagnosis.

Thirty-three subjects satisfied the inclusion criteria. CT assessment for visual semantic features and computer-aided texture analysis was performed on non-contrast CT scan obtained immediately prior to the biopsy. All scans were performed on Siemens Definition CT scanners (Munich, Germany), at 120 kV, with a dose of 90 mAs, using a 256-slice scanner with 1 mm collimation, and reconstruction at 1 mm slice thickness and B30f (medium kernel) reconstruction. The CT-detected nodules underwent visual semantic feature assessment independently by a thoracic and interventional radiologist with 23 years of experience (FA) and a thoracic radiologist with 6 years of experience (SMHT), with any discrepancies resolved through consensus. The readers were blind to the demographic data and corresponding pathology results associated with CT images. A total of 23 semantic variables were identified and assessed using CT scans obtained immediately prior to the biopsy. These features are described in Table 1.

CAD-based texture analysis of nodules for characterization was performed on the same baseline non-contrast CT scan obtained before the biopsy with 1 mm slice thickness at B30f reconstruction. The CT images were imported to an in-house-developed software package, the Quantitative Imaging Workstation (QIWS) developed at UCLA for segmentation and further image analysis. This workstation has a variety of automated and semi-automated CAD and measurement tools. Automatic segmentation function was applied slice-by-slice for each nodule by an operator trained at identifying and segmenting nodules (SJ). Each slice was then carefully reviewed, and manual corrections to the segmentation were made as necessary. Thereafter, a radiologist with 23 years of experience (FA) reviewed the segmentation, performed necessary segmentation adjustments, and finalized the segmentation. A sample of the segmentation is shown in Figure 1. Quantitative features were extracted from the regions of interest using the QIWS software (Version 1), as previously mentioned. These features included shape, intensity histogram statistics, and texture features derived from gray-level co-occurrence matrices (GLCM). Through CAD-based texture analysis, 95 variables were generated, categorized into 11 histogram features, 4 size features, and 80 GLCM texture features (as detailed in the Appendix A). The radiomics calculations followed the Image Biomarker Standardization Initiative (IBSI) guidelines [12].

### 2.2. Statistical Analysis

Summary statistics of visual read and radiomic variables were reported for invasive vs. noninvasive LPA (AIS/MIA). The association between invasiveness of adenocarcinoma and visual read were analyzed using Fisher’s exact test or Chi-squared test, depending on the counts of the groups. We used LASSO for feature selection based on a cross-validation approach, with mean deviance as the criterion. Given the nature of difference in visual read and texture feature as well as extensive number of variables, the 23 categorical variables from visual read underwent LASSO selection in three groups: nodule internal features, nodule external features, and features representing anatomical location; resulting in 7 visual read variables selected. The 95 continuous variables based on texture feature extraction were composed of 3 datasets: histogram, size, and GLCM. These are shown in Appendix A. Analysis was performed using R version 4.1.1 and STATA 17.0 SE.

## 3. Results

Out of the 33 patients with CT-guided biopsy diagnosis of AIS/MIA who satisfied the eligibility criteria and were selected for the analysis, the final post-surgery pathologic assessment of the explanted lesion was consistent with AIS/MIA in 9 (27.3%) cases, and invasive LPA in 24 (72.7%) cases. The study population consisted of 20 (60.6%) females and 13 (39.4%) males, with an average age of 69.8 (±10.6) years. A summary of the patients’ demographics is provided in Table 2. Examples of the CT images immediately before biopsy are depicted in Figure 2 and Figure 3.

### 3.1. Visual Read Variables

On CT, visual semantic features included 21 (63.6%) part-solid, 5 (15.2%) pure ground glass, and 7 (21.2%) solid nodules. From the LASSO selection method (assessing the independent vs. dependent variables), four semantic feature variables were identified for the model but were not statistically significant. These variables included septal stretching, nodule surround, secondary margin type, and small cyst-like spaces. Summary statistics of the visual read variables are shown in Table 3. After subjecting the variables to another round of LASSO selection and backward selection, no variables or combination of variables were found to be statistically significant in differentiating iLPA from AIS/MIA. Multivariable exact logistic regression from visual semantic feature variables selected from forward selection is shown in Table 4.

### 3.2. Radiomic Variables

For radiomic features, 95 variables were assessed in the categories of histogram data, size data, and feature data. Three variables were selected by forward selection: Tumor_perc95 (95th percentile, HU), volume, and glcm_info_corr_a_16′ (linear dependency of gray level values to the respective voxels in GLCM). The summary statistics for the radiomic variables are shown in the Appendix A. Multivariable logistic regression results from radiomic feature variables selected from forward selection are shown in Table 5.

## 4. Discussion

In this study, we intend to predict iLPA using visual and CAD-based texture analysis on nodules which were initially diagnosed as MIA or AIS based on CT-guided biopsy. The purpose of core needle biopsy is to characterize the nodule and within adenocarcinoma spectrum predict the probability of a lung nodule being invasive or non-invasive. The accurate diagnosis of invasiveness status of adenocarcinoma carries significant prognostication and helps guide treatment plans as well as plan future interventions in case biopsy results are discordant [13]. LPA is well defined and has been incorporated in the lung cancer staging TNM 8th edition [14]. The predictors and imaging features of LPA have also been established in the literature. Lepidic growth is characterized by ground glass nodules on CT imaging, and the degree of concurrent solid component can suggest invasiveness, with MIA lesions having equal or less than 5 mm and invasive LPA lesions having more than 5 mm of solid tissue on imaging. The literature has also suggested that semantic and radiomic features may be useful in the process of differentiating invasive from noninvasive LPA [15,16]. In a study involving a Caucasian cohort, univariate analysis showed nodule height, solid component size, density, mass, disappearance rate, and pleural retraction were found to be significant differentiating factors between AIS/MIA and invasive disease. On multivariate analysis, only the solid component size was significant [14].

There are multiple diseases that can mimic LPA on CT scans and clinicians rely mostly on the results of CT-guided biopsy to determine the course of care for these LPAs [17]. However, as demonstrated here, core biopsy alone cannot capture the heterogeneity of the entire tumor. The implications of these findings are far reaching as there will be patients who may later be upstaged due to incomplete biopsy findings that suggest AIS/MIA disease. In this study, we demonstrate that the majority (24/33—72.7%) of the patients who only had noninvasive tumor on the core biopsy, suggesting AIS/MIA, were later found to be iLPA, on the explant specimen. Prior studies have also demonstrated concordance between core biopsy and final resection pathologies ranging between 58.6% and 77% in lepidic adenocarcinoma [18,19]. In another study looking at peripheral subsolid nodules, the overall concordance rate between biopsy and surgical pathology in determining the predominant histological subtype was 64%. There was better concordance for tumors less than 2 cm or pure GG nodules [20]. Therefore, patients and clinicians who receive biopsy results suggestive of AIS/MIA disease should anticipate the probability of more advanced disease on the explant specimen. This can lead to more appropriate surgical management to minimize further complications and evolution of malignancy. It is worth noting that recent studies have attempted to develop models for differentiating and predicting the invasiveness of lung adenocarcinoma based on specific patterns observed on chest CT, with promising results. For example, Yang et al. trained a nomogram incorporating intratumoral and peritumoral radiomics features from CT scans, combined with clinical semantic data, to predict poorly differentiated invasive pulmonary adenocarcinoma. Their study demonstrated that this nomogram has the potential to preoperatively predict poorly differentiated IPA manifesting as subsolid or solid lesions [21]. Similarly, Chen et al. developed a multi-parameter prediction model integrating monochromatic CT values from dual-energy CT (DECT) along with quantitative and semantic features, which showed promise in distinguishing invasive lung adenocarcinoma from AIS and MIA in GG-predominant lung adenocarcinomas [22].

Although multiple semantic parameters did not reach statistical significance, predictions can be made when assessing nodules for concordance with biopsy results. Some semantic features showed potential for predicting invasiveness, with septal stretching absent in all noninvasive cases but present in six (25%) of the invasive cases. In addition, solid consistency was not observed in any of the non-invasive samples, whereas it was noted in seven (29.2%) of the invasive cases.

In our study, we used both visual and CAD-based texture analysis to predict invasive LPA and increase the concordance between CNB and surgical specimen. Using the LASSO technique, four variables were selected from the visual assessments: secondary margin type, small cyst-like spaces, nodule surround emphysema, and septal stretching. Using the forward selection method, three variables were identified from the radiomic feature variables: Tumor_Perc95, Volume, and glcm_info_corr_a_16. Although these seven variables were identified to be predictive, they were not able to exceed a statistically significant level of *p* < 0.15 after they were subjected to another LASSO selection and backward selection. Tumor_Perc95 represents the 95th percentile of the Hounsfield unit of pixels within the margins of the tumor. Glcm_cor or Gray Level Co-occurrence Matrix is a statistical texture analysis method that observes the probability of a pair of pixels occurring.

Limitations to this study included a relatively small sample size with an unbalanced dataset in an 8:3 (invasive–AIS/MIA) ratio. This may explain the inability to precipitate more semantic variables that were predictive of invasiveness. Selection bias could also be an issue in our study, as patients who received resection despite AIS/MIA on biopsy may have had other predictors of lung cancer beyond imaging at the time of the procedure including growth, family history of cancer, smoking history, or other factors increasing lung cancer risk. These factors may have also prompted the surgeon to resect the tumor as opposed to pursuing non-operative management such as Stereotactic Body Radiation Therapy or Image-Guided Thermal Ablation. Future studies with standardized reconstruction algorithms and higher sample sizes may be able to address these limitations and provide more robust findings to validate the importance of visual semantic and radiomic features. In addition, we acknowledge that achieving statistical significance at the conventional threshold of *p* < 0.05 can be challenging due to the limited sample size. Given the 96 radiomic features in our dataset, we decided to use a strict feature selection criterion to minimize the risk of overfitting. However, we recognize that this approach may not be optimal and could potentially exclude features with weaker, yet clinically relevant, associations.

## 5. Conclusions

A large number of patients with noninvasive lepidic pattern lung adenocarcinoma on needle biopsy are later found to have invasive tumor on surgical specimen. Although statistically non-significant, some semantic features showed potential for predicting invasiveness, with septal stretching absent in all noninvasive cases, and solid consistency present in a significant portion of invasive cases.

## Figures and Tables

**Figure 1 medsci-12-00057-f001:**
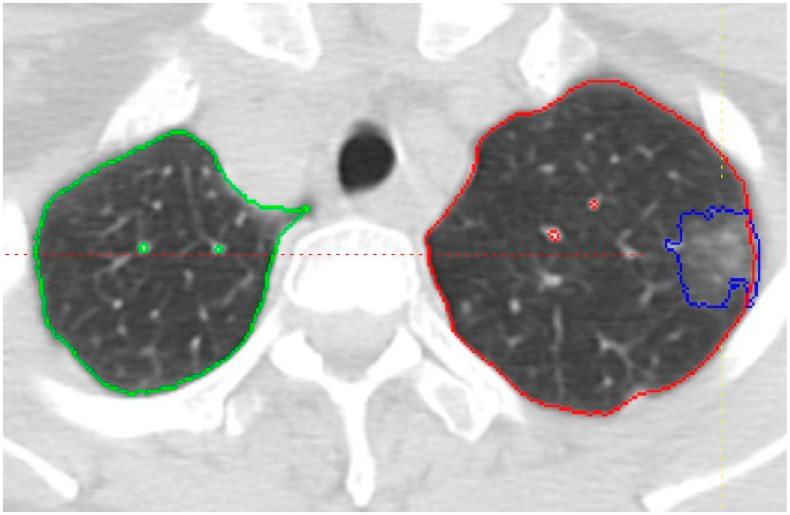
Axial CT image showing the segmentation of a part-solid nodule in the left upper lobe, outlined in blue. The right and left lungs are outlined in green and red, respectively. Vessels are segmented separately to distinguish them from the lung parenchyma.

**Figure 2 medsci-12-00057-f002:**
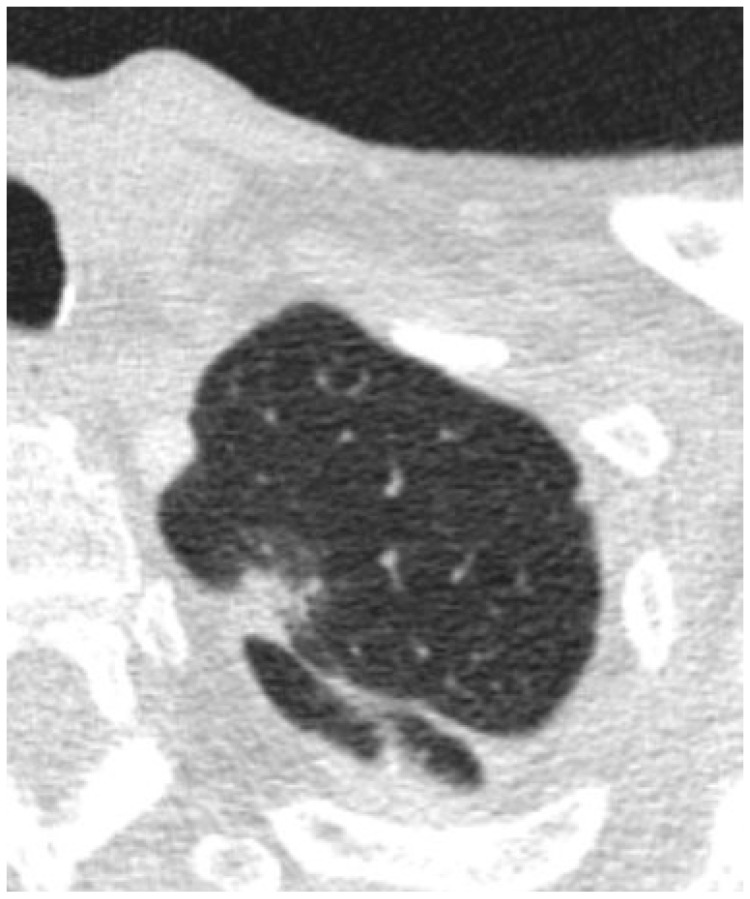
Axial images depicting a nodule that was found to be AIS/MIA after resection.

**Figure 3 medsci-12-00057-f003:**
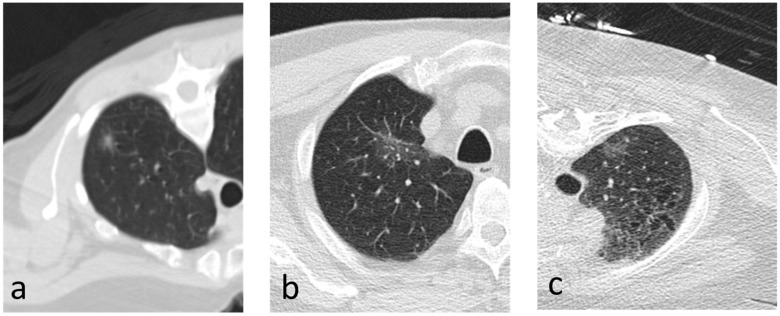
(**a**–**c**) Axial images depicting nodules that were found to be invasive on resection.

**Table 1 medsci-12-00057-t001:** Description of visual semantic features.

Variable	Description
Intranodular Bronchiectasis Availability of prior scan Lobar location Longest axial diameters Longest diameter of solid component Nodule consistency Nodule margin conspicuity Nodule reticulation Nodule shape Purely endobronchial Short axial diameter Subpleural	As implied by the variable name.
Airway cut-off	An airway entering the nodule is obliterated at some point after entering the nodule.
Axial location	Peripheral: if located within 1 cm of costal pleura. Central: if not peripheral.
Cyst-like spaces	The presence of cystic spaces smaller than 3 mm within the nodule’s border.
Cavitation	The presence of cystic spaces with thick walls and a size of 3 mm or larger within the nodule’s borders.
Primary dominant margin	The nodule’s most dominant margin type.
Secondary dominant margin	The second most dominant type of nodule margins.
Paracicatricial emphysema	Some or all of the lung immediately surrounding the nodule is fibrotic and emphysematous.
Nodule surround emphysema	Emphysema is the nature of lung parenchyma in the 25 mm surrounding environment.
Vascular convergence	A vessel that approaches or departs from the nodule appears to curve or deviate from its anticipated course in order to connect with the nodule.
Pleural attachment	The nodule is attached to the costal pleura.
Pleural retraction	The nodule pulls the adjacent pleura so that a dimpling is created.
Septal stretching	Septal lines without tapering are observed in the parenchyma surrounding the nodule (as opposed to spiculations that taper as going away from the nodule).

**Table 2 medsci-12-00057-t002:** Summary of the patients’ demographics.

Demographic	Category	Count (Percent)
Sex	Female	20 (60.6)
	Male	13 (39.4)
Age	Under 50	1 (3.0)
	50–59	1 (3.0)
	60–69	14 (42.4)
	70–79	13 (39.4)
	80 and over	4 (12.1)
Smoking History	Yes	20 (60.6)
	No	13 (39.4)
Race	Asian	8 (24.2)
	Caucasian	21 (63.6)
	Black	1 (3.0)
	Other	3 (9.1)

**Table 3 medsci-12-00057-t003:** Summary statistics of visual semantic features read.

Characteristics	Non-Invasive *n* = 9	Invasive *n* = 24	Total *n* = 33	*p*-Value
Airway Cut-off, N (%)				1.000
No	9 (100)	23 (95.8)	32 (97.0)	
Yes	0	1 (4.2)	1 (3.0)	
Axial Location, N (%)				1.000
Central (>1 cm)	2 (22.2)	6 (25.0)	8 (24.2)	
Peripheral (≤1 cm)	7 (77.8)	18 (75.0)	25 (75.8)	
Cavitation, N (%)				1.000
No	9 (100.0)	22 (91.7)	31 (93.9)	
Yes	0 (0)	2 (8.3)	2 (6.1)	
Intranodular Bronchiectasis, N (%)			2 (6.1)	0.097
No	9 (100.0)	17 (70.8)	7 (21.2)	
Yes	0 (0)	7 (29.2)	24 (72.7)	
Large Pericystic Space, N (%)				1.000
No	9 (100)	23 (95.8)	32 (97.0)	
Yes	0 (0)	1 (4.2)	1 (3.0)	
Lobar Location, N (%)				0.335
LLL	0 (0)	5 (20.8)	5 (15.2)	
LUL	4 (44.4)	6 (25.0)	10 (30.3)	
RLL	1 (11.1)	3 (12.5)	4 (12.1)	
RML	0 (0)	0 (0)	0 (0)	
RUL	4 (44.4)	10 (41.7)	14 (42.4)	
Nodule Consistency, N (%)				0.015
Part-solid	6 (66.7)	15 (62.5)	21 (63.6)	
Pure Ground Glass	3 (33.3)	2 (8.3)	5 (15.2)	
Solid	0 (0)	7 (29.2)	7 (21.2)	
Nodule Margin Conspicuity, N (%)				0.681
Poorly Marginated	7 (77.8)	15 (62.5)	22 (66.7)	
Well Marginated	2 (22.2)	9 (37.5)	11 (33.3)	
Nodule Reticulation, N (%)				0.295
No	7 (77.8)	22 (91.7)	29 (87.9)	
Yes	2 (22.2)	2 (8.3)	4 (12.1)	
Nodule Shape, N (%)				0.688
Complex	5 (55.6)	17 (70.8)	22 (66.7)	
Ovoid	2 (22.2)	3 (12.5)	5 (15.2)	
Round	2 (22.2)	4 (16.7)	6 (18.2)	
Nodule Surround (Approx. 2.5 cm)				0.174
Emphysema	2 (22.2)	1 (4.2)	3 (9.1)	
Normal	7 (77.8)	23 (95.8)	30 (90.9)	
Paracicatricial Emphysema				-
No	9 (100)	24 (100)	33 (100)	
Yes	0 (0)	0 (0)	0 (0)	
Pleural Attachment				0.358
No	3 (33.3)	4 (16.7)	7 (21.2)	
Yes	6 (66.7)	20 (83.3)	26 (78.8)	
Pleural Retraction				0.315
Absent	4 (44.4)	7 (29.2)	11 (33.3)	
Mild Dimpling	2 (22.2)	11 (45.8)	13 (39.4)	
Obvious Dimpling	0	1 (4.2)	1 (3.0)	
n/a	3 (33.3)	5 (20.8)	8 (24.2)	
Primary Dominant Margin				0.135
Indeterminate	8 (88.9)	14 (58.3)	22 (66.7)	
Lobulated	0 (0)	6 (25.0)	6 (18.2)	
Notched/Concavity	0 (0)	0 (0)	0 (0)	
Smooth	1 (11.1)	1 (4.2)	2 (6.1)	
Spiculated/Serrated	0 (0)	3 (12.5)	3 (9.1)	
Purely Endobronchial				-
No	9 (100)	24 (100)	33 (100)	
Yes	0 (0)	0 (0)	0 (0)	
Secondary Margin Type				0.229
Intermediate	9 (100)	15 (62.5)	24 (72.7)	
Lobulated	0 (0)	4 (16.7)	4 (12.1)	
Notched/Concavity	0 (0)	1 (4.2)	1 (3.0)	
Smooth	0 (0)	0 (0)	0 (0)	
Spiculated/Serrated	0 (0)	4 (16.7)	4 (12.1)	
Septal Stretching				0.090
No	9 (100)	18 (75.0)	27 (81.8)	
Yes	0 (0)	6 (25.0)	6 (18.2)	
Small Cyst-like Spaces				0.107
No	8 (88.9)	13 (54.2)	21 (63.6)	
Yes	1 (11.1)	11 (45.8)	12 (36.4)	
Subpleural				-
No	0 (0)	4 (16.7)	4 (12.1)	
Yes	0 (0)	1 (4.2)	1 (3.0)	
n/a	9 (100)	21 (87.5)	28 (84.9))	
Vascular Convergence				1.000
No	9 (100)	23 (95.8)	32 (97.0)	
Yes	0 (0)	1 (4.2)	1 (3.0)	

**Table 4 medsci-12-00057-t004:** Multivariable exact logistic regression from visual semantic feature variables selected from forward selection.

Visual Variable	Odds Ratio	95% CI	*p*-Value
Septal Stretching	2.30	0.33–infinity	0.25
Nodule Surround (Approx 2.5 cm)			
Emphysema	0.38	0–3.36	0.23
Normal	reference		
Secondary Margin Type			
Lobulated	1.40	0.17–infinity	0.40
Notched/Concavity	0.25	0.013–infinity	0.80
Spiculated/Serrated	1.95	0.27–infinity	0.30
Indeterminate	reference		
Small Cyst-like Spaces	2.16	0.13–137.50	0.97

**Table 5 medsci-12-00057-t005:** Multivariable logistic regression results from radiomic feature variables selected from forward selection.

Radiometric Feature Variable	Odds Ratio	95% CI	*p*-Value
Tumor_perc95	1.00	1.00–1.01	0.15
Volume	1.00	1.000–1.001	0.22
glcm_info_corr_a_16	0.02	0.00013–2.9	0.124

## Data Availability

Data supporting the findings of this study are available from the corresponding author upon reasonable request. Due to privacy and ethical restrictions, the dataset are not publicly available.

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
