# Peer review of "Predicting Invasiveness in Lepidic Pattern Adenocarcinoma of Lung: Analysis of Visual Semantic and Radiomic Features"

_medsci, 2024, doi:10.3390/medsci12040057_

Round 1
Reviewer 1 Report
Comments and Suggestions for Authors
Congratulations to you on your brilliant study on how to increase the accuracy of the pre-operative prediction of invasiveness of pulmonary malignancy. The entire manuscript was well written with good language and layout. Actually there are no obvious defect of the study and analysis and I believe the visual semantic and quantitative CAD-based texture analysis is a good method, using radiomics, for improving the diagnosis. However, the study is a retrospective one and the results are statistical not significant. I suggest you can conduct a prospective study of ten to twenty cases, implementing your new diagnostic strategy, in order to verify its accuracy. This result can be added to the present one and re-submit as a new artcle.
Author Response
Congratulations to you on your brilliant study on how to increase the accuracy of the pre-operative prediction of invasiveness of pulmonary malignancy. The entire manuscript was well written with good language and layout. Actually there are no obvious defect of the study and analysis and I believe the visual semantic and quantitative CAD-based texture analysis is a good method, using radiomics, for improving the diagnosis. However, the study is a retrospective one and the results are statistical not significant. I suggest you can conduct a prospective study of ten to twenty cases, implementing your new diagnostic strategy, in order to verify its accuracy. This result can be added to the present one and re-submit as a new article.
Response: This is an excellent recommendation, and we will certainly consider it for future research.
Reviewer 2 Report
Comments and Suggestions for Authors
The manuscript of "Predicting Invasiveness in Lepidic Pattern Adenocarcinoma of Lung: Analysis of Visual Semantic and Radiomic Features " by Drs. Johnson, et al reports the results of utilizing visual semantic and computer-aided detection (CAD)-based texture features to differentiate the AIS or MIA with CT-guided biopsy. A total of 33 patients with 24 had invasive LPA and 9 had AIS/MIA were included in the analysis. Least absolute shrinkage and selection operator (LASSO) method or forward variable selection was used to select the most predictive feature and combination of semantic and texture features for detection of invasive lung adenocarcinoma. nodules. LASSO method selected seven variables for the model, but all were not statistically significant. “Volume” was found to be statistically significant when assessing the correlation between independent variables using the backward selection technique. The LASSO selected “tumor_Perc95”, “nodule surround”, “small cyst-like spaces” and “volume” when assessing the correlation between independent variables. They conclude that although statistically non-significant, some semantic features showed potential for predicting invasiveness, with septal stretching absent in all noninvasive cases, and solid consistency present in a significant portion of invasive cases.
Overall, the manuscript was well prepared. Here are some concerns on the analysis and report of the data.
1. Variables selection and model development: Due to limited sample size, the usual statistical significance (p-value < 5%) is hard to select features unless very strong correlation.
2. For LASSO features selection, what is criteria used for variable selection, Deviance Residuals or classification errors? For the binary outcome, usually classification errors are used for features selection. Thus, it is the classification error rate, not the statistical significance should be presented.
Author Response
Thank you so much for your insightful comments and suggestions. Below, please find a point-by-point response to the reviewers’ comments.
Overall, the manuscript was well prepared. Here are some concerns on the analysis and report of the data.
1- Variables selection and model development: Due to limited sample size, the usual statistical significance (p-value < 5%) is hard to select features unless very strong correlation.
Response: thank you for raising this important point. We acknowledge that achieving statistical significance at the conventional threshold of p < 0.05 can be challenging due to the limited sample size. Given the 96 radiomic features in our dataset, we decided to use a strict feature selection criterion to minimize the risk of overfitting. However, we recognize that this approach may not be optimal and could potentially exclude features with weaker, yet clinically relevant, associations. We have added this point to our limitations.
2- For LASSO features selection, what is criteria used for variable selection, Deviance Residuals or classification errors? For the binary outcome, usually classification errors are used for features selection. Thus, it is the classification error rate, not the statistical significance should be presented.
Response: We used LASSO for feature selection based on a cross-validation approach, with mean deviance as the criterion. Our goal was to identify important visual and radiomic features associated with the invasiveness of adenocarcinoma, rather than building a classification model. Given the extensive number of variables and the limited sample size, mean deviance was chosen as a more appropriate criterion for inference in this context. In the future when large dataset is available, we agree to use the criteria of classification error rate to build a classifier. We have added an explanation to the Materials and Methods under the section 2.2 to clarify this.
Reviewer 3 Report
Comments and Suggestions for Authors
The authors retrospectively studied the semantic and radiomic features of CT to identify invasive cancers from AIS and MIA biopsy results. The sample size was small and the results were very preliminary. Although most of the features were disappointing, the authors found some semantic features with the potential to predict invasiveness. Their results may be clinical relavent. However, many studies have been conducted in similar designs (e.g., Front Oncol. 2024 Jan 19;14:1289555. or Acad Radiol. 2024 Jul;31(7):2962-2972.). My comments and suggestions are listed as follows,
1. The abstract seems a little bit lengthy. Please consider revising it to be more concise.
2. In the "Materials and Methods," the authors may need to disclose more details of CT protocol, e.g., tube voltage, current, and reconstruction matrices.
3. In line 115-119, the authors described their semi-automatic segmentation methods. Please explain in more detail how you segment lesions. Perhaps with a Figure to explain.
4. The authors may need to disclose their software for radiomic feature calculation. For example, were they using Pyradiomics or LIFEx? Or other software. Also, did their radiomics calculation follow the IBSI definition?
5. If available, please summarize patient demographics in a Table.
6. In Table 2, Nodule Consistency seemed to be the only semantic feature that was significantly associated with invasive adenocarcinoma. If logistic regression is conducted with Nodule Consistency, will it show a better predictive value than the other four semantic features in Table 3?
7. In Table 4, multivariable logistic regression of radiomics features did not show a statistically significant association with invasiveness. How about the combination of the three features?
8. In the Introduction, the authors mentioned that some demographics, such as smoking history, were associated with invasive disease. Did these demographics also show an association with invasive disease in your study?
9. This article has a Supplementary file. However, the Supplementary file seems missing.
Author Response
Thank you so much for your insightful comments and suggestions. Below, please find a point-by-point response to the reviewers’ comments.
- The abstract seems a little bit lengthy. Please consider revising it to be more concise.
Response: the initial abstract was 262 words. We have revised it for conciseness, and it is now 242 words.
- In the "Materials and Methods," the authors may need to disclose more details of CT protocol, e.g., tube voltage, current, and reconstruction matrices.
Response: We have added these details to the Materials and Methods section.
- In line 115-119, the authors described their semi-automatic segmentation methods. Please explain in more detail how you segment lesions. Perhaps with a Figure to explain.
Response: we have added explanation about the segmentation process and the software used for it. We have also added Figure 3 to illustrate a sample of our nodule segmentation.
- The authors may need to disclose their software for radiomic feature calculation. For example, were they using Pyradiomics or LIFEx? Or other software. Also, did their radiomics calculation follow the IBSI definition?
Response: we have included details on the software used for radiomics feature extraction and analysis in the Materials and Methods section. Additionally, the radiomics calculations followed the IBSI guidelines, and this has been clarified in the Materials and Methods section.
- If available, please summarize patient demographics in a Table.
Response: we have added table 5 to summarize the patients’ demographics, providing additional details.
- In Table 2, Nodule Consistency seemed to be the only semantic feature that was significantly associated with invasive adenocarcinoma. If logistic regression is conducted with Nodule Consistency, will it show a better predictive value than the other four semantic features in Table 3?
Response: that is really excellent point. If we used the entire dataset, nodule consistency would be the best fit in model. In our semantic feature dataset, we used cross validation. Due to the lack of sample size and unbalanced invasive versus non-invasive adenocarcinoma, nodule consistency was not selected during the cross validation lasso feature selection, as stated in the Results section of the manuscript.
- In Table 4, multivariable logistic regression of radiomics features did not show a statistically significant association with invasiveness. How about the combination of the three features?
Response: that’s excellent point, we could use factor analysis or principle component analysis to combine the three features. Instead, we divide the radiomic feature into 3 components (historgram, size, GLCM driven features) and used the top 3 features from each model to show the multivariable model. Again, our goal was to understand the radiomic features with respect to invasiveness (increase odds with the size of volume and intensity and decrease odds in spatial correlation), not trying to build the model due to limited sample size.
- In the Introduction, the authors mentioned that some demographics, such as smoking history, were associated with invasive disease. Did these demographics also show an association with invasive disease in your study?
Response: Thank you for the insightful suggestion. Due to the small size of our cohort and the fact that it was not matched for demographic factors, it was not feasible to perform a meaningful analysis of demographic data, such as smoking history, in relation to invasive disease. However, we have added a new table to summarize the patients’ demographic data.
- This article has a Supplementary file. However, the Supplementary file seems missing.
Response: we apologize for this oversight. We have uploaded the supplementary material with the revision submission.
Round 2
Reviewer 1 Report
Comments and Suggestions for Authors
Since you have not added the information I suggested, I cannot accept the manuscript for publication.
Author Response
Since you have not added the information I suggested, I cannot accept the manuscript for publication.
Response: Our research was a pilot retrospective analysis, intended as a foundational step for future studies. We hope the findings will support securing funding for further work. While a prospective study would be a valuable extension, it requires additional resources and is beyond the scope of the current study, which aims to provide preliminary insights into the utility of semantic and CAD-based texture analysis in predicting the invasiveness of lepidic-pattern pulmonary adenocarcinoma. We value your recommendation and intend to explore it in future studies, as it would certainly enhance the diagnostic accuracy of this approach.
Reviewer 2 Report
Comments and Suggestions for Authors
As Authors stated, the key of the analysis is to see the concordance of statistical model predicted result with actual data. The multivariable analysis presented there do not provide with much useful information to readers. It would be helpful for Authors to present the concordance table from the multivariable analysis instead of the table 3, and 4 there.
Author Response
As Authors stated, the key of the analysis is to see the concordance of statistical model predicted result with actual data. The multivariable analysis presented there do not provide with much useful information to readers. It would be helpful for Authors to present the concordance table from the multivariable analysis instead of the table 3, and 4 there.
Response: Thank you for your suggestion. We understand the importance of presenting a concordance table to illustrate the performance of the multivariable analysis. However, we opted to use mean deviance criteria, which is a conservative scoring rule for evaluating predictive models (Friedman J, Hastie T, Tibshirani R. Regularization paths for generalized linear models via coordinate descent. Journal of statistical software. 2010;33(1):1). In addition, to better evaluate the performance of visual features, we have included the area under the curve (AUC) from ROC analyses for each feature. We believe this combination provides a more sensitive assessment than the classification rate approach. However, we will consider incorporating a concordance table in future studies to enhance clarity.
Reviewer 3 Report
Comments and Suggestions for Authors
The authors have addressed my comments and suggestions. However, modeling could not be done due to the limited sample size. Perhaps the modeling can be done in the authors' future works.
Author Response
The authors have addressed my comments and suggestions. However, modeling could not be done due to the limited sample size. Perhaps the modeling can be done in the authors' future works.
Response: We agree that modeling is limited by the current sample size and plan to explore it further in future studies with a larger dataset. While a prospective study would be a valuable extension, it requires additional resources and is beyond the scope of the current study, which aims to provide preliminary insights into the utility of semantic and CAD-based texture analysis in predicting the invasiveness of lepidic-pattern pulmonary adenocarcinoma.